# Evidence for Maintained Post-Encoding Memory Consolidation Across the Adult Lifespan Revealed by Network Complexity

**DOI:** 10.3390/e21111072

**Published:** 2019-11-01

**Authors:** Ian M. McDonough, Sarah K. Letang, Hillary B. Erwin, Rajesh K. Kana

**Affiliations:** Department of Psychology, The University of Alabama, Tuscaloosa, AL 35487, USA; sletang@crimson.ua.edu (S.K.L.); hberwin@crimson.ua.edu (H.B.E.); rkkana@ua.edu (R.K.K.)

**Keywords:** aging, consolidation, default mode network, episodic memory, fMRI, multiscale entropy, network complexity, resting state

## Abstract

Memory consolidation is well known to occur during sleep, but might start immediately after encoding new information while awake. While consolidation processes are important across the lifespan, they may be even more important to maintain memory functioning in old age. We tested whether a novel measure of information processing known as network complexity might be sensitive to post-encoding consolidation mechanisms in a sample of young, middle-aged, and older adults. Network complexity was calculated by assessing the irregularity of brain signals within a network over time using multiscale entropy. To capture post-encoding mechanisms, network complexity was estimated using functional magnetic resonance imaging (fMRI) during rest before and after encoding of picture pairs, and subtracted between the two rest periods. Participants received a five-alternative-choice memory test to assess associative memory performance. Results indicated that aging was associated with an increase in network complexity from pre- to post-encoding in the default mode network (DMN). Increases in network complexity in the DMN also were associated with better subsequent memory across all age groups. These findings suggest that network complexity is sensitive to post-encoding consolidation mechanisms that enhance memory performance. These post-encoding mechanisms may represent a pathway to support memory performance in the face of overall memory declines.

## 1. Introduction

As people age, experiencing memory decline is a common occurrence [1]. According to the associative deficit hypothesis [2], these age-related decreases in episodic memory are due to weakened abilities to encode simple, unrelated units of information together into a more complex unit (i.e., associating a picture of a scene with a picture of a face), and to retrieve that complex unit. The ability to associate and bind features together has been shown to be mediated by the hippocampus (for reviews, see [3,4]). Despite these clear links, a meta-analysis conducted on fMRI studies investigating age differences in successful and unsuccessful memory encoding revealed overall stability in hippocampal functioning in old age [5]. This finding suggests that, under some circumstances, the hippocampus can be successfully recruited to aid memory performance across the adult lifespan. In the present study, we investigated the degree to which a novel measure of information processing might be sensitive to key episodic memory mechanisms within the hippocampus and associated brain regions in a sample of young, middle-aged, and older adults.

One mechanism that might contribute to age-related memory declines is consolidation. The most well-characterized consolidation mechanisms occur during slow-wave sleep and rapid eye movement stages of sleep via neural replay and changes in synaptic strengths [6,7,8]. These processes occur between hippocampal and neocortical regions to help create enduring episodic contexts and promote the generalization of semantic representations [9]. Moreover, it has been well documented that sleep becomes more disrupted with increasing age, including disruptions in slow-wave sleep and rapid eye movement sleep [10,11]. These age differences have been shown to reduce the benefits of sleep for cognition normally found in young adults [12].

More recently, accumulating evidence in animals has suggested that similar neural replay processes related to consolidation occur while awake soon after new memories are encoded [13]. In humans, fMRI has been used to support post-encoding consolidation mechanisms in an awake state [14,15,16]. In some of these studies, resting-state scans were collected in young adults before and after memory encoding to assess changes in functional connectivity between the hippocampus and the neocortex. Across these studies, functional connectivity increased from pre-encoding to post-encoding, and this increase has been associated with better subsequent memory performance. Together, these findings have been interpreted as evidence for systems-level consolidation in an awake state, at least in young adults. 

Only a few studies have investigated post-encoding consolidation in older adults [17,18,19]. Of these aging studies, only one directly tested whether changes in functional connectivity within the default mode network (DMN) before and after memory encoding differed with age [17]. Interestingly, they did not find age differences in pre-post change in DMN connectivity, but rather found greater age-related changes in connectivity within the salience network and reduced age-related changes in connectivity within the occipito-temporal network. In addition to finding these age-related differences in non-DMN networks, they found that the coupling between the DMN and executive function network decreased after encoding for older adults and increased for younger adults. Furthermore, the decreased coupling between the two networks was associated with better memory performance only in older adults. The authors interpreted this finding as the need for older adults to suppress interference from competing networks during post-encoding consolidation. While this interpretation remains a possibility, measures other than functional connectivity may capture unique aspects of neural replay mechanisms that unfold over time between the hippocampus and neocortical regions. 

Multiscale entropy (MSE) is one such measure that might capture novel properties of the brain over time. This method of analysis estimates the complexity of a physiological time series using temporal coarse-graining procedures to evaluate signals at multiple temporal scales, in recognition of the likelihood that the dynamic complexity of biological signals might operate across a range of temporal scales [20,21]. Across these different temporal scales, the degree of randomness (i.e., entropy) is estimated by searching for repeated patterns of small temporal segments. To the extent that repeated patterns are found, then the physiological signal is quantified as having lower entropy and is interpreted as containing less unique information. In contrast, researchers have argued that higher entropy across temporal scales is associated with richer information [22,23] or more integrated information [24,25]. For example, research has found that brain entropy increases when retrieving new episodic information compared with known semantic information [26]. More recently, applying repetitive transcranial magnetic stimulation to create a “virtual lesion” in the frontal cortex reduced brain entropy [27], further suggesting that greater entropy is associated with more information processing. Notably, greater entropy is not simply interpreted as randomness that is equivalent to noise. When entropy is estimated across temporal scales, clear differences in patterns emerge [28,29]. For example, many types of noise decrease as temporal scales increase, whereas entropy showed dynamic patterns of increases from fine to mid temporal scales followed by slow declines as the temporal scales became coarser [28]. Thus, this analysis is often referred to as “complex” because of these dynamic changes in entropy and deviation from patterns of noise.

Other evidence for the usefulness of quantifying the complexity of brain signals comes from neuropsychiatric disorders. Studies have found that aberrant functional connectivity is a hallmark feature of many disorders, such as Alzheimer’s disease [30], schizophrenia [31], autism spectrum disorder (ASD) [32,33], attention deficit hyperactivity disorder [34], and mood disorders [35]. Nevertheless, traditional functional connectivity analyses have fallen short of explaining these syndromes comprehensively, primarily due to their neural heterogeneity. To fill this gap, MSE analyses have been applied to investigate neuropsychiatric disorders and have found differences depending on both temporal scale and brain region. Using electroencephalography (EEG), neural complexity has been related to observation/imitation tasks in children with ASD [36], severity of ASD [37], and has been used to predict autism and risk for autism with relatively high accuracy [38]. Using fMRI, neural complexity has been shown to differ in patients with schizophrenia, bipolar disorder, and schizoaffective disorder compared to healthy control individuals [39,40,41,42,43,44,45,46,47,48,49,50].

From an aging perspective, complexity has been proposed to decrease across many physiological systems [41]. Specifically, EEG and magnetoencephalography (MEG) studies have found decreases in neural complexity in mild cognitive impairment and Alzheimer’s disease [42]. Decreases in neural complexity in old age and in disorders might underlie the deficits in selective cognitive processing in these conditions. Using fMRI during rest, decreased neural complexity has been reliably found in various regions across the brain in older adults [29,43,44,45,46], and these decreases have sometimes been related to poorer cognition [46]. However, one study highlighted some of the limitations of using MSE analyses to estimate neural complexity in fMRI in comparison to MEG [47]. In this study, neural complexity decreased with age when using both fMRI and MEG. However, the temporal precision of MEG allowed for a greater number of temporal scales, revealing additional relationships with hypoperfusion surrounding neuronal damage due to stroke. To the extent that more time points are collected, however, more temporal scales can be estimated [28,48].

### Participants

Here, we capitalized on the well-established finding that temporal patterns cluster across different sets of brain regions to form intrinsic connectivity networks [49,50,51]. Such similar temporal patterns suggest that the complexity of those patterns also would be similar within each brain network (at least at rest), allowing us to estimate network complexity [28,48] involved in memory consolidation. Specifically, we aimed to test the extent that network complexity in the DMN might be used as a novel measure of post-encoding consolidation mechanisms across the adult lifespan. To the extent that network complexity is sensitive to and a proxy for information processing within memory networks, we predicted that an increase in network complexity from pre-encoding to post-encoding would be associated with memory performance for the task administered in the fMRI session. To the extent that the well-known age-related deficits in episodic memory for recollected details are due to impaired consolidation mechanisms, then we also should see an age-related decline in such post-encoding processes as measured by network complexity. We focused on the changes in network complexity within the DMN because of the high degree of connectivity between the DMN and the hippocampus and the known relationships between the two for consolidation.

## 2. Materials and Methods 

### 2.1. Participants

Participants were drawn from the Alabama Brain Study on Risk for Dementia. Details from the study can be found in our earlier publication assessing dementia risk and brain activity during memory retrieval [52]. All participants were recruited from the Tuscaloosa and Birmingham areas within Alabama through word of mouth, flyers, Facebook ads, and newsletters. Participants were excluded if they had contra-indicators for magnetic resonance imaging (MRI), were left-handed, had a prior diagnosis of any neurological condition, stroke, traumatic brain injury, claustrophobia, or history of substance abuse. Young adults aged 20–30 were recruited from the local community to serve as a baseline group. Middle-aged and older adults ranging in age from 50 to 74 were included if they were free of dementia as measured by the St. Louis University Mental Status (SLUMS) [53], spoke English fluently, were right-handed, and had at least one of the following self-reported risks for dementia: subjective memory complaints, less than a high school education, African American or Hispanic ethnoracial category, mild head trauma, family history of Alzheimer’s disease, current diagnosis of hypertension or systolic blood pressure greater than 140 mmHg, current diagnosis or a family history of heart disease, current diagnosis of high total cholesterol, history or current use of smoking tobacco, current diagnosis or family history of diabetes, and body mass index greater than 30 kg/m^2^. All participants gave informed consent using methods approved by the institutional review board at the University of Alabama. Participants’ vision was normal or corrected to normal using MR-compatible glasses or contact lenses. Demographic characteristics can be found in Table 1. 

### 2.2. Procedures

Across two sessions, participants completed cognitive and MRI batteries. From the cognitive battery, we used the scaled word reading subtest from the Wide Range Achievement Test-4 to control for premorbid IQ in all analyses. Note that one participant’s premorbid IQ score was missing and was imputed using a regression-based matching technique [52]. The MRI session included scans in the following order: resting-state, memory encoding, T1-structural scan, resting-state, memory retrieval, and a visual-motor checkerboard task. The analyses here focus on the resting-state scans that occurred before and after the memory encoding phase. 

### 2.3. fMRI Scans

The two resting-state scans consisted of 175 volumes over 5 min each. Participants were told to close their eyes but not fall asleep. After the first resting-state scan, participants studied 64 pairs of pictures for the memory encoding task over two 8-minute sessions. The pictures consisted of either a face-object or face-scene pair for 3 s. After viewing each pair, participants were given 2.16 s to predict how likely they would remember the pair on a later memory test on a 3-point scale corresponding to likely, maybe, or unlikely. The inter-trial interval ranged from 1.72 to 17.20 s. The memory test consisted of a five-alternative-choice test in which participants were presented with a previously seen face and were asked to choose the object or scene that was previously paired with the face from five options: two objects, two scenes, and “never seen.” Of the four possible picture choices, one was the target and three were lures. Because all options were previously seen, participants needed to rely on recollection processes to answer correctly. The “never seen” option specifically referred to not remembering the face, thus precluding participants from making a correct response without complete guessing. Each of the 64 memory trials was presented for 5.16 s, with intertrial intervals that ranged from 1.72 to 10.32 s. The memory test was divided into two runs lasting for 5 min. 

### 2.4. fMRI Acquisition and Preprocessing

A 3T Siemens PRISMA scanner at the UAB Civitan International Neuroimaging Laboratory was used to collect MRI scans. High resolution T1-weighted structural MPRAGE scans were acquired using (parallel acquisition acceleration type = GRAPPA; acceleration factor = 3, TR = 5000 ms, TE = 2.93 ms, TI 1 = 700 ms, TI 2 = 2030 ms, flip angle 1 = 4°, flip angle 2 = 5°, FOV = 256 mm, matrix = 240 × 256 mm^2^, in-plane resolution = 1.0 × 1.0 mm^2^). All functional scans used T2*-weighted EPI sequences (56 interleaved axial slices, 2.5 mm thickness, TR = 1720 ms, TE = 35.8 ms, flip angle = 73°, FOV = 260 mm, matrix = 104 × 104 mm, in-plane resolution = 2.5 × 2.5 mm^2^, multi-band acceleration factor = 4). 

The functional data were unwarped, coregistered to the structural scan, and spatially smoothed (8 mm FWHM kernel) using Statistical Parametric Mapping 12 (SPM12). The blood oxygen level dependent (BOLD) signal was then denoised using Multivariate Exploratory Linear Optimized Decomposition into Independent Components (MELODIC) [54]. The resulting spatiotemporal components were flagged using an in-house script that applied machine learning to frequency and temporal elements indicative of potential artifacts. The flagged components were then regressed from the BOLD signal, also using MELODIC. The denoised data were then warped into a study template using Advanced Neuroimaging Tools (ANTs) [55]. 

### 2.5. Resting-state fMRI Analysis

Dual regression analyses using FMRIB Software Library (FSL)’s “dual_regression” function [56,57] were implemented to isolate the time series within 10 major resting-state networks (RSNs) [51] to estimate subject-specific functional connectivity patterns within each of the networks. This method was used to extract the single time series common across all voxels within each network, which can then be used to estimate network complexity [28]. The template for the networks was transformed into the space from our sample template using ANTS. To implement dual regression, each RSN template is used as a spatial predictor for each subject’s denoised 4D BOLD data in the first general linear model (GLM) regression. This regression is used to find the best matching time course for a given RSN. In a second regression, the resulting time course from the first regression is used as a set of temporal regressors in a GLM to estimate individual regression weights in the spatial domain, which represents the degree to which a time series in each voxel matches the time series for that component. The output from these dual regressions is a subject and network-specific time series along with a spatial Z-scored map.

### 2.6. MultiScale Entropy (MSE) Analysis

Network complexity was calculated by computing MSE on each of the time series that were created from the dual regression analysis. MSE estimates sample entropy across multiple temporal scales [20,21]. Different temporal scales are created by averaging neighboring time points within non-overlapping windows. This process is repeated to create a new coarse-grained time series that captures neural dynamics at different levels. Sample entropy is separately estimated for each of the created time series. We estimated seven temporal scales due to the length of the resting-state scans. Previous work has shown that the variability of MSE increases as the temporal scale increases, leading to unreliable estimates in the BOLD signal [28,48]. We used the heuristic of dividing the number of time points (*N* = 175) by 25 for our upper bound number of temporal scales to use. Sample entropy is defined as the natural logarithm of the conditional probability that a given pattern of data of a specified length (*m*) repeats at the next time point for the entire time series at a given scale factor (of a dataset with a total length *N*). It considers subsequent patterns to be a repeat of the given pattern if they match within a certain tolerance (*r*) such that larger tolerance values increase the number of matches [58,59]. To the extent that a time series has a greater number of pattern matches, the time series is less random, and the entropy value is lower. In contrast, a smaller number of pattern matches is characterized as being more random, yielding a greater entropy value. We selected our parameters based on those used in prior studies investigating MSE using fMRI: *m* = 2 and *r* = 0.5 [28,29,48,60]. 

### 2.7. Multilevel Modeling Analyses

Given the nested nature of the temporal scales within a given network for each subject, multilevel modeling (MLM) was used to test the effects of MSE differences as a function of age and memory performance. The lme4 package in R was used for data analysis [61]. The MLM analyses used a random intercept and random slope for Timescale, modeled an auto-correlation structure of 1, and used maximum likelihood estimation. MSE was modeled at the first level and Timescale at the second level. Thus, each of the seven temporal scales were modeled simultaneously in the analyses. Adding an interaction term in the analyses with Timescale would indicate that the effects differed by Timescale. However, none of our primary analyses revealed significant interactions with Timescale, indicating that the relationships among MSE and the factors of interest (e.g., age and memory accuracy) did not depend on Timescale. For simplicity, the models with the Timescale interaction terms are not reported. All MLM analyses controlled for pre-encoding network complexity, sex (male = 0, female = 1), and premorbid IQ. 

Note that our sample was recruited to enrich risk factors for dementia across middle-aged and older adults. Because the young adults were not assessed for dementia risk, a covariate across all participants could not be calculated in the primary analyses. Instead, we created a dementia risk score in the middle-aged and older adults only by summing the presence of any of the dementia risk inclusion criteria (see Section 2.1). We then reconducted the MLM analysis in the middle-aged and older adults with the dementia risk score in the model (Model 4) to determine (1) if dementia risk modifies the post-encoding network complexity in the DMN, and (2) if the inclusion of the dementia risk score modifies the effects of aging or memory accuracy on post-encoding complexity.

Although theories of consolidation propose that the hippocampus closely coordinates with regions within the DMN to successfully store information, networks that process object representations also might be involved in post-encoding consolidation [16]. Additionally, to the extent that the post-encoding network complexity is due to covert rehearsal (i.e., re-visualization) of the study information following encoding, then we might expect similar changes in network complexity in attention or cognitive control networks. To test these possibilities, we conducted additional MLM analyses investigating the change in network complexity in four additional networks [51]: lateral occipito-temporal network (Model 5), cingulo-opercular network (Model 6), left fronto-parietal network (Model 7), and right fronto-parietal network (Model 8).

## 3. Results

Table 1 summarizes the descriptive characteristics among the three age groups. The biggest differences between the groups were in relation to cognition. A one-way analysis of variance (ANOVA) indicated that associative memory performance in the fMRI task differed between groups, (*F* (2,83) = 24.55, *p* < 0.001) such that memory for young adults was significantly greater than that of middle-aged (*p* < 0.001) and older adults (*p* < 0.001). Memory performance did not differ between middle-aged and older adults (*p* = 0.40). In addition, premorbid IQ differed with age (*F* (2,83) = 6.20, *p* = 0.003) such that middle-aged adults scored significantly poorer than young adults (*p* = 0.028) and older adults (*p* = 0.004), who did not differ from one another (*p* = 0.98). The proportion of self-reported belonging to a particular ethnoracial category also differed between age groups (*χ*^2^ (4) = 22.66, *p* = 0.001).

The dual regressions successfully captured the 10 RSNs [51]. Of particular interest was the DMN both pre and post encoding. Notably, the hippocampus also was strongly correlated with the DMN in our sample (Figure 1). Thus, the resulting time series spanned both the DMN and bilateral hippocampi. As shown in Figure 2, network complexity increased slightly across the seven temporal scales, but these increases were not significant at pre-encoding rest or post-encoding rest (*p*’s > 0.20). Additionally, mean network complexity in the DMN did not differ between pre- and post-encoding (*p* = 0.71), but they were correlated with one another (*r* = 0.13, *p* = 0.001). 

In the MLM analysis, our primary predictors were Age and Memory Accuracy nested within Timescale, and our dependent variable was the difference in Network Complexity with the DMN between pre-encoding and post-encoding (Model 1). Both older age (*p* = 0.0088) and higher memory accuracy (*p* = 0.043) were associated with a greater increase in network complexity from pre- to post-encoding (Figure 3). The full results from the MLM analysis can be found in Table 2. An additional test for an Age × Memory Accuracy interaction was not significant (*p* = 0.61). Although we found evidence consistent with our hypothesis that network complexity increased following the encoding of information, difference scores can be driven by effects at pre-encoding only, post-encoding only, or both. Thus, we conducted two additional tests to ensure that there were no pre-existing differences before memory encoding (Model 2) and to verify that the network complexity difference score was primarily due to differences at post-encoding (Model 3). We found no effect of Age or Memory Accuracy on Network Complexity during pre-encoding (*p* > 0.59). However, we did find significant effects of Age (*p* = 0.010) and Memory Accuracy (*p* = 0.041) on Network Complexity during post-encoding (see Table 2). Thus, the locus of the network complexity difference score appears to be isolated to changes in MSE at post-encoding.

To assess whether controlling for dementia risk would alter the findings, we conducted additional sensitivity analyses excluding the younger adults. We found that, even in this smaller sample, older age (*p* = 0.040) and memory accuracy (*p* = 0.011) were associated with a larger network complexity change in the DMN (see Table 2). In addition, the effect sizes for age and memory accuracy nearly doubled after young adults were removed from the analyses. No significant association was found for dementia risk (*p* = 0.44).

Finally, to test the extent that other networks also were involved in post-encoding consolidation [16,62] or involved covert rehearsal (i.e., re-visualization) of the study information following encoding, we conducted additional MLM analyses in four additional networks [51]: lateral occipito-temporal network (Model 5), cingulo-opercular network (Model 6), left fronto-parietal network (Model 7), and right fronto-parietal network (Model 8). The results from these analyses can be found in Table 2 In short, neither Age nor Memory Accuracy were significantly related to change in Network Complexity across any of the other RSNs (*p*’s > 0.13).

## 4. Discussion

The present study aimed to test the degree to which network complexity would be sensitive to post-encoding consolidation mechanisms across the adult lifespan. This study is the first to our knowledge to assess post-encoding consolidation mechanisms using network complexity in fMRI. We predicted that network complexity within the DMN would increase following memory encoding, that this change in network complexity would be correlated with subsequent memory performance, and that changes in network complexity would partially explain age-related declines in episodic memory. Across participants, network complexity did not increase following memory encoding. However, substantial individual differences existed such that a greater increase in network complexity from pre- to post-encoding in the DMN was associated with better memory performance, as predicted. Although this change in network complexity was associated with age, the direction of the effect was contrary to our predictions. Older age was associated with greater increases in network complexity. We elaborate on these main findings below.

Greater increases in network complexity predicted better memory performance, suggesting that this measure successfully captured portions of post-encoding consolidation processes. This finding converges with work showing that the intensity of reactivation during neural replay at rest is associated with better memory performance in rats [63]. We also found that these effects were limited to the DMN. Importantly, the DMN captured by our dual regression analyses encompassed bilateral hippocampi, suggesting that the complexity in the BOLD signal may represent interactions between the hippocampus and portions of the neocortex that included the medial prefrontal cortex and the medial parietal cortex. Prior research in both human and animal models has established that consolidation-related stabilization of memories relies on the connections between the hippocampus and the neocortex, supporting the use of our network-based analyses [64,65]. Moreover, brain regions within the DMN specifically are well-known for supporting episodic memory (for review, see [66]). Indeed, studies using resting-state functional connectivity have found that reduced connectivity between the hippocampus and other parts of the DMN is associated with poorer memory in older adults [67,68]. 

We did not find the same evidence for other brain networks that might aid in suppressing competing information during post-encoding consolidation via cognitive control networks [17] or that might aid in consolidating perceptual information in the occipito-temporal network [14,17,69]. One reason we may not have found evidence for the involvement of these other networks is that specific brain regions involved in processing faces (e.g., fusiform gyrus) or places (e.g., parahippocampal gyrus) may have shown subtle alterations in MSE that were not captured by our relatively broader network approach. Another consideration is that our sample consisted of mostly middle-aged and older adults who show age-related declines in specialized processing of visual information [70,71,72]. Such reductions in visual-information processing may have reduced the ability of these regions to contribute substantially to post-encoding consolidation mechanisms. Supporting this idea, age-related decreases in pre- and post-encoding functional connectivity have been found in the occipito-temporal network [17]. 

Why might network complexity capture post-encoding consolidation? Current evidence in animal models suggests that neural replay occurs in a temporally reversed order from how the encoded events originally unfolded [13]. Therefore, the temporal patterns of brain activity, as assessed via network complexity, should provide insightful information into consolidation processes. Additionally, neural replay is thought to unfold more rapidly than the original event and repeated many times, suggesting that new and different information is coded within brain networks mediating consolidation processes [13]. To the extent that network complexity represents greater information processing offline and can capture neural dynamics at multiple time scales, then network complexity should be sensitive to consolidation processes. 

In the context of the positive correlations we found between network complexity and memory performance, we were surprised to find that aging also was accompanied by increases in network complexity within the DMN. Older age was not associated with changes in network complexity in the other networks, suggesting that this aging effect was specific to the DMN and not a general physiological effect of the aging brain. One intriguing possibility for these unexpected age effects is that the increases in network complexity represent a novel compensatory mechanism to help ameliorate age-related memory declines. Some cognitive aging theories propose that much of the age-related deficits in memory occur because of reduced processing resources that limits their ability to initiate effortful strategies [73,74]. This deficit in controlled processing also is believed to be the origin of the enhanced reliance on automatic memory processes with age [75,76]. The present findings suggest that post-encoding consolidation processes might be considered an automatic process that is not only intact, but also helps maintain cognition in older age. 

A slightly different perspective is that older adults may rely more on processes subserved by the DMN generally and, depending on the context, this over-reliance might aid or hinder memory [77]. Indeed, older adults often rely more on prior knowledge and schemas than younger adults [78]. Processing such schemas has been associated with connectivity between the medial temporal lobes and brain regions within the DMN, such as the medial prefrontal cortex in similar post-encoding paradigm, at least in young adults [16,79]. Thus, the present findings are consistent with the notion that middle-aged and older adults rely more on the DMN than younger adults [77].

In contrast to these optimistic perspectives, the present findings stand in contrast to two recent studies that assessed MSE using EEG in a similar rest–task–rest paradigm, but that did not measure episodic memory [80,81]. For example, MSE has been found to increase at fine temporal scales in anterior regions and decrease at coarse temporal scales in posterior brain regions from pre- to post-task in young adults [81]. In that study, no differences in MSE were found from pre- to post-task in middle-aged or older adults. Similar reductions in old age were found during a visual stimulation task [80]. Any differences between studies could easily be attributed to differences between EEG and fMRI. However, we also believe that their findings were sensitive to a different process altogether. In [81], the positive correlations between MSE at rest with behavioral performance before and after the task (separately) were similar from fine to mid temporal scales. Because their study was not measuring episodic memory (but rather perception and working memory), the authors did not need to assess correlations between pre- to post-task changes in MSE and behavioral performance as we did in the current study. Thus, while any task can be considered an “encoding” task because information is constantly being encoded, an alternative hypothesis is that individual differences in resting MSE might provide a reliable individual difference estimate for cognitive health [81]. Our findings appear to be different from this perspective because individual differences in network complexity before encoding were not related to either age or memory performance as it was in their study. The specificity of our results suggests that our analyses are not capturing general individual differences in cognitive health, but specific post-encoding factors.

However, another possibility is that brain activity during rest after encoding represents the recovery and restoration of the brain following an effortful task, especially in the DMN [19,82,83,84,85]. On the one hand, the increased network complexity with age might be interpreted as an increase in “effort” with age to restore the brain to a pre-task state. This explanation would be broadly in line with theories proposing age-related deficits in neural efficiency [86]. Of course, such “effort” would have to be interpreted in a non-traditional manner, given that one does not have conscious awareness of brain processes occurring at rest. A related argument is that increases in network complexity relate to overt rehearsal of the encoding material in preparation for the later memory test. Indeed, such overt rehearsal would support later memory performance. Several reasons make this explanation unlikely. First, attempting to rehearse faces verbally can lead to later memory impairments known as verbal overshadowing [87,88]. Second, the encoding phase consisted of 64 face-picture pairs, which would be difficult to rehearse. Lastly, a structural scan was collected immediately after the last encoding scan. Therefore, any effects of rehearsal would likely have come to an end or at least would have been minimized by the time the second resting-state scan began.

## 5. Conclusions

Network complexity in the DMN during post-encoding awake states may represent a novel assessment of memory consolidation. We showed that pre- to post-change in network complexity related to better subsequent memory performance, supporting the idea that greater network complexity represents the degree of information processing in the brain [28,48]. This change in network complexity also increased with age, possibly representing a mechanism for older adults to maintain memory in the face of overall age-related declines in episodic memory. Network complexity and other measures of brain entropy have quickly become adopted as novel measures to assess optimal brain functioning, but more research is needed to understand further how these measures relate to more established brain measures such as task activations and how to interpret different temporal scales. Additionally, future work is needed to replicate and provide supporting evidence for this novel mechanism underlying post-encoding mechanisms of memory, including how network complexity might be related to neural replay.

## Figures and Tables

**Figure 1 entropy-21-01072-f001:**
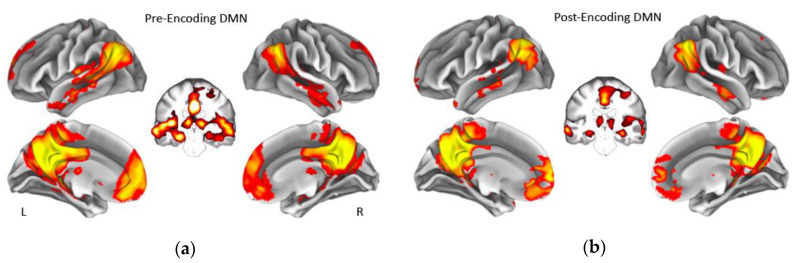
Group average of the default mode network during rest before memory encoding (**a**) and during rest after memory encoding (**b**). Coronal slices show that the default mode network in our sample also included bilateral hippocampus before and after encoding. Displayed maps were thresholded at a Z-score > 7. L = Left, R = Right. DMN = default mode network.

**Figure 2 entropy-21-01072-f002:**
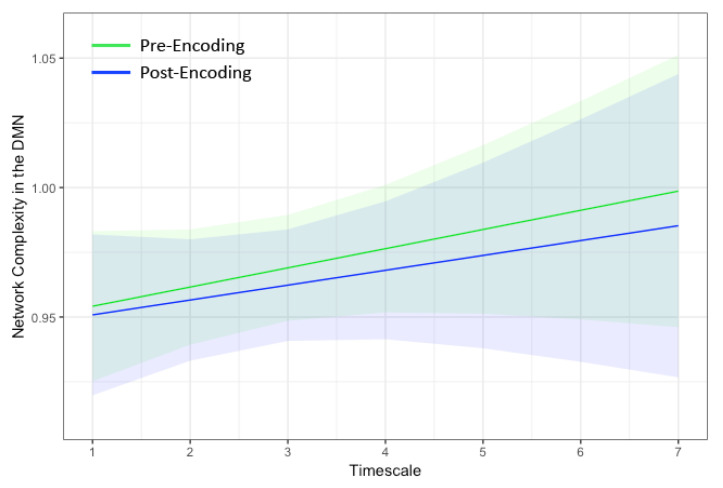
Marginal plots showing network complexity in the DMN estimated using multiscale entropy (MSE) on the y-axis at the seven temporal scales within the default mode network on the x-axis during rest before encoding (green) and after encoding (blue). The shaded regions represent 95% confidence intervals. Network complexity increased linearly with temporal scale and did not differ between pre- and post-encoding rest. DMN = default mode network.

**Figure 3 entropy-21-01072-f003:**
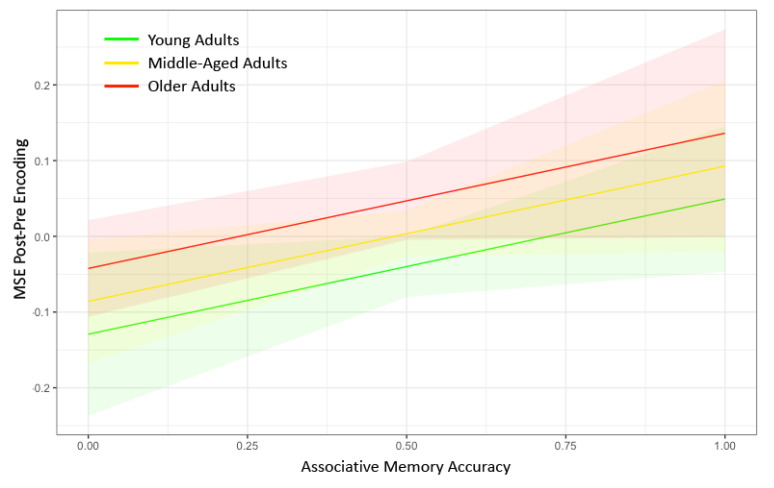
Marginal plots showing the relationship between post–pre encoding network complexity in the default mode network on the y-axis and associative memory performance on the x-axis for younger adults (green), middle-aged adults (yellow), and older adults (red). The shaded regions represent 95% confidence intervals. Greater associative memory performance was associated with greater increases in network complexity in the default mode network for each of the age groups. Older age was associated with greater increases in network complexity in the default mode network. No interactions were found between age and memory performance on change in network complexity. MSE = multiscale entropy.

**Table 1 entropy-21-01072-t001:** Participant Characteristics.

	Young Adults	Middle-Aged Adults	Older Adults	Group Differences
N	20	31	35	-
Mean Age	23.35 (3.25)	54.29 (2.84)	66.17 (4.02)	*F* (2,83) = 987.70, *p* < 0.001
Age Range	20–30	50–60	61–74	-
Sex (F/M)	11 (55%)/9 (45%)	17 (54%)/14 (46%)	23 (66%)/12 (34%)	*χ*^2^ (2) = 1.01, *p* = 0.61
Ethnoracial Category				*χ*^2^ (4) = 22.66, *p* < 0.001
Non-Hispanic White	12 (60%)	14 (45%)	28 (80%)	-
African American	1 (5%)	13 (42%)	7 (20%)	-
Other	7 (35%)	4 (13%)	0 (0%)	-
Years of Education	15.00 (2.20)	14.26 (2.67)	13.83 (2.93)	*F* (2,83) = 1.20, *p* = 0.30
SLUMS Score	-	26.48 (2.95) ^1^	26.21 (2.91)	*t* (62.7) = 0.38, *p* = 0.70
Associative Memory Performance	0.57 (0.17)	0.36 (0.14)	0.32 (0.09)	*F* (2,83) = 24.55, *p* < 0.001
Premorbid IQ	106.65 (10.69)	95.16 (16.08)	107.55 (16.70)	*F* (2,83) = 6.20, *p* = 0.003
Dementia Risk	-	5.32 (1.85)	5.03 (2.08)	*t* (62.9) = 0.66, *p* = 0.51

^1^ Missing score for one participant. SLUMS = St. Louis University Mental Status.

**Table 2 entropy-21-01072-t002:** Fixed Effects for Network Complexity Multilevel Modeling Analyses.

	DMN Post–Pre Encoding (Model 1)	DMN Pre-Encoding (Model 2)	DMN Post-Encoding (Model 3)	DMN Post–Pre Encoding (Model 4)	OT Post–Pre Encoding (Model 5)	CO Post–Pre Encoding (Model 6)	LFP Post–Pre Encoding (Model 7)	RFP Post–Pre Encoding (Model 8)
Intercept	−0.0084 (0.013)	0.9764 (0.012) ***	0.9683 (0.013) ***	−0.0242 (0.024)	−0.0051 (0.016)	−0.0162 (0.017)	−0.0674 (0.015) ***	−0.0745 (0.017) ***
Timescale	0.0099 (0.013)	0.0148 (0.012)	0.0117 (0.013)	0.0065 (0.015)	−0.0010 (0.006)	0.0021 (0.007)	0.0175 (0.006) **	0.0156 (0.007) *
MSE Pre-Encoding	−0.1789 (0.009) ***	-	-	−0.1813 (0.010) ***	−0.1836 (0.009) ***	−0.1800 (0.009) ***	−0.1913 (0.009) ***	−0.1992 (0.009) ***
Age	0.0376 (0.014) **	−0.0002 (0.014)	0.0374 (0.014) *	0.0750 (0.036) *	0.0017 (0.016)	−0.0155 (0.015)	−0.0155 (0.014)	−0.0039 (0.015)
Sex (Ref. = Male)	−0.0036 (0.010)	0.0096 (0.010)	−0.0027 (0.011)	−0.0034 (0.012)	0.0014 (0.012)	0.0021 (0.011)	0.0104 (0.011)	0.0032 (0.011)
Premorbid IQ	−0.0090 (0.013)	0.0015 (0.013)	−0.0084 (0.013)	−0.0222 (0.015)	0.0123 (0.014)	−0.0055 (0.014)	−0.0020 (0.013)	−0.0074 (0.014)
Memory Accuracy	0.0321 (0.016) *	0.0083 (0.016)	0.0329 (0.016) *	0.0538 (0.021) **	0.0156 (0.018)	−0.0039 (0.017)	−0.0238 (0.016)	0.0144 (0.017)
Dementia Risk	-	-	-	0.0052 (0.007)	-	-	-	-

*** *p* < 0.001, ** *p* < 0.01, * *p* <0.05; Values represent standardized beta coefficients and their standard errors in parentheses; MSE = Multiscale entropy; DMN = Default mode network; OT = Occipito-temporal Network; CO = Cingulo-opercular network; LFP = Left fronto-parietal network; RFP = Right fronto-parietal network.

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
