# Peer review of "Evidence for Maintained Post-Encoding Memory Consolidation Across the Adult Lifespan Revealed by Network Complexity"

_entropy, 2019, doi:10.3390/e21111072_

Round 1

Reviewer 1 Report

The authors proposed the use of multiscale entropy (MSE) for study and analysis of the post-encoding consolidation mechanisms in a sample of young, middle-aged, and older adults. The authors reported that“aging was associated with an increase in network complexity from pre- to post-encoding in the default mode network (DMN)” and that suchincreases in network complexity in the DMN also were associated with better subsequent memory across all age groups.”

Major Comments:

1) It is not clear how the authors used the MSE values in different scales. Did they utilize their average? Did they use them at all scales but then fit the regression to all these values? Please provide more clear explanation on how MSEs at different scales were used in your analyses.

2) Line 307 “this change in network complexity would be correlated with subsequent memory” and Line 315, “Greater increases in network complexity predicted better memory performance, suggesting that this measure successfully captured portions of post-encoding consolidation processes”: It would be useful and informative if the authors also provide the result of correlation analysis between pre- and post-encoding to better assess the observed change in MSE. In fact, to better realize the nature of such a potential relation, the authors might want to consider non-linear measures such as mutual information. Subsequently, they might also want to look into conditional entropy between the pre- and post-encoding MSEs and determine (for example using bootstrapping) whether the observed increase in post-encoding (i.e., after accounting for pre-encoding through conditioning) is significant (e.g., bootstrap at 95% confidence interval (CI) with zero not falling within this CI and that the average bootstrap result > 0 i.e., a significant increase in MSE from pre- to post-encoding).

3) Lines 402-404, “Network complexity also increased with age, possibly representing a mechanism for older adults to maintain memory in the face of overall age-related declines in episodic memory performance.”:MSEhas been shown to havehigher valuesin the finer time scales, accompanied with their decrease in the coarser scales. This isattributedto a shift from long–range brain regions’ connections (captured by coarse scales MSEs) to a more local processing by aging (e.g., [25] and [28] in the authors’ manuscript).It would be informative if the authors can provide some explanation/interpretation, comparing their findings on increased complexity in the older adults and these other results that are related with change in complexity (as captured in different MSE scales) with aging.

Minor Comments:

1) Line 191: “an equation is used to estimate individual regression”: It is more informative if the authors can provide more details about this equation. Have they computed a closed-form themselves? Was it estimated based on some approximation such as gradient descent? Did they use GLM for calculating this equation? What were the dependent and independent variables for estimating/calculating this equation? etc.

2) Lines 270-278 and lines 290-298: This information should be presented in the Statistical Analysis Section and not the Results Section.

3) Lines 320-322, “Additionally, neural replay is thought to unfold more rapidly than the original event and repeated many times, suggesting that new and different information is coded within brain networks mediating consolidation processes.”: This part requires proper citations from the literature.

Author Response

We thank Reviewer 1 for their comments. We have attempted to respond to each of them below and we feel the manuscript is much clearer. The original points made by the reviewer are in italics with our responses immediately after in non-italics.

Reviewer 1

The authors proposed the use of multiscale entropy (MSE) for study and analysis of the post-encoding consolidation mechanisms in a sample of young, middle-aged, and older adults. The authors reported that “aging was associated with an increase in network complexity from pre- to post-encoding in the default mode network (DMN)” and that such “increases in network complexity in the DMN also were associated with better subsequent memory across all age groups.”

Major Comments:

1) It is not clear how the authors used the MSE values in different scales. Did they utilize their average? Did they use them at all scales but then fit the regression to all these values? Please provide more clear explanation on how MSEs at different scales were used in your analyses.

MSE was nested within Timescale in our multilevel model analyses. In this way, all Timescales were modeled simultaneously in the model. Adding an interaction term in the analysis with Timescale would indicate that the effects differed by Timescale. However, none of our primary analyses revealed significant interactions with Timescale, thus indicating that the relationships among MSE and the factors of interest (e.g., age, memory accuracy) did not depend on Timescale. We have now clarified this in the manuscript.

2) Line 307 “this change in network complexity would be correlated with subsequent memory” and Line 315, “Greater increases in network complexity predicted better memory performance, suggesting that this measure successfully captured portions of post-encoding consolidation processes”: It would be useful and informative if the authors also provide the result of correlation analysis between pre- and post-encoding to better assess the observed change in MSE. In fact, to better realize the nature of such a potential relation, the authors might want to consider non-linear measures such as mutual information. Subsequently, they might also want to look into conditional entropy between the pre- and post-encoding MSEs and determine (for example using bootstrapping) whether the observed increase in post-encoding (i.e., after accounting for pre-encoding through conditioning) is significant (e.g., bootstrap at 95% confidence interval (CI) with zero not falling within this CI and that the average bootstrap result > 0 i.e., a significant increase in MSE from pre- to post-encoding).

We have now included the correlation between pre- and post-encoding MSE (r = .13, p = .001). Although significant, the weak correlation suggests that there is not a strong relationship between them. Moreover, MLM analyses are actually well-suited to accurately parse within-subject variance and between-subject variance (in contrast to least squares general linear models), thus serving as an appropriate analytic method. Because pre-encoding MSE was included in the model, we are achieving the same goal as what the reviewer suggests by calculating conditional entropy (i.e., estimating the change in MSE while accounting for pre-encoding MSE). The intercept term in each of the model actually shows a similar effect to what the reviewer is proposing; that is, after controlling for the other variables in the model, a significant intercept assesses whether the mean difference (post-pre) in MSE is significantly different from zero. In the case of Model 1 (using the DMN), this mean difference does not significantly differ from zero. However, we note that we are not interested in this mean difference per se, but rather what factors predict the variance in the change in MSE.

3) Lines 402-404, “Network complexity also increased with age, possibly representing a mechanism for older adults to maintain memory in the face of overall age-related declines in episodic memory performance.”: MSE has been shown to have higher values in the finer time scales, accompanied with their decrease in the coarser scales. This is attributed to a shift from long–range brain regions’ connections (captured by coarse scales MSEs) to a more local processing by aging (e.g., [25] and [28] in the authors’ manuscript).It would be informative if the authors can provide some explanation/interpretation, comparing their findings on increased complexity in the older adults and these other results that are related with change in complexity (as captured in different MSE scales) with aging.

The quote was actually a typo, so we thank the reviewer for catching this. What we meant was that the change in network complexity also increased with age. As the reviewer points out, while there have been many previous studies investigating differences in level of MSE with age (at fine and coarse time scales), very little research has investigated pre-post changes while controlling for these mean level age differences, thus precluding our ability to thoroughly compare our findings with other studies. The only two studies that we are aware of that tested changes in MSE in older adults were the Wang et al. (2016) and Takahashi et al. (2009) papers using EEG (not fMRI). In the Discussion, we compare our findings with those prior two papers. Because we did not find interactions with temporal scale (see earlier comment), we cannot speak to differences in fine, mid, and coarse grained scales. Moreover, because fMRI is limited in the precision in temporal scales, we hesitate to directly compare temporal scales to MSE using EEG.

Minor Comments:

1) Line 191: “an equation is used to estimate individual regression”: It is more informative if the authors can provide more details about this equation. Have they computed a closed-form themselves? Was it estimated based on some approximation such as gradient descent? Did they use GLM for calculating this equation? What were the dependent and independent variables for estimating/calculating this equation? etc.

We have attempted to clarify how the dual regressions are calculated. First, we did not create the equations ourselves, but used FSL’s “dual_regression” function. Second, the regression equations relied on GLM based models (which is the norm for most fMRI analyses). The first regression used the spatial map as the predictor and fMRI data as the dependent variable. The second regression used the time course as the predictor and fMRI data as the dependent variable. We refer the reviewer to the FSL method papers for a more detailed explanation (Beckmann, C.F.; Mackay, C.E.; Filippini, N.; Smith, S.M. Group comparison of resting-state FMRI data using multi-subject ICA and dual regression. Neuroimage 2009, 47, S148; Filippini, N.; MacIntosh, B.J.; Hough, M.G.; Goodwin, G.M.; Frisoni, G.B.; Smith, S.M.; Matthews, P.M.; Beckmann, C.F.; Mackay, C.E. Distinct patterns of brain activity in young carriers of the APOE-ε4 allele. Proc. Natl. Acad. Sci. U.S.A. 2009, 106, 7209-7214.).

Our paragraph now states, “Dual regression analyses using FSL’s “dual_regression” function [56,57] were implemented to isolate the time series within 10 major RSNs from [51] to estimate subject-specific functional connectivity within each of the networks. This method was used to extract the single time series common across all voxels within each network, which can then be used to estimate network complexity [28]. The template for the networks was transformed into the space from our sample template using ANTS. To implement dual regression, each RSN template is used as a spatial predictor for each subject’s denoised 4D BOLD data in the first general linear model (GLM) regression. This regression is used to find the best matching time course for a given RSN. In a second regression, the resulting time course from the first regression is used as a set of temporal regressors in a GLM to estimate individual regression weights in the spatial domain, which represents the degree to which a time series in each voxel matches the time series for that component. The output from these dual regressions is a subject and network-specific time series along with a spatial Z-scored map.”

2) Lines 270-278 and lines 290-298: This information should be presented in the Statistical Analysis Section and not the Results Section.

We have moved the rationale and descriptions of the statistical analyses in the Statistical Analysis Section as recommended. However, we still kept a brief summary of the goal of the new analyses in the Results section to remind the readers why we were conducting the other analyses.

3) Lines 320-322, “Additionally, neural replay is thought to unfold more rapidly than the original event and repeated many times, suggesting that new and different information is coded within brain networks mediating consolidation processes.”: This part requires proper citations from the literature.

We have added a citation here.

Reviewer 2 Report

Comments to authors are given in the attached file 'Entropy review'

Author Response

We thank Reviewer 2 for their comments. We have attempted to respond to each of them below and we feel the manuscript is much clearer. The original points made by the reviewer are in italics with our responses immediately after in non-italics.

Reviewer 2

The article provides new data on age-related differences in the brain states associated with memory encoding. In particular the authors propose that the well-known decrements in memory performance related to aging may be attributable to decrements in post-encoding consolidation. Accordingly they measured changes in network complexity in the DMN between pre- and post-encoding resting states by means of multiscale entropy measures. In fact “mean network complexity did not differ between pre- and post-encoding” (lines 240-241) but nonetheless greater MSE increases between these phases related to successful memory performance and also (surprisingly) to participant age – that is, older participants showed greater increases in network complexity from pre- to post-encoding phases than did their younger counterparts.

This seems to me (as a cognitive psychologist as opposed to a neuroscientist) to be a well-run and well-analyzed study on an important current topic – the neural bases of age-related memory loss. The main puzzling finding (acknowledged by the authors) is that whereas memory success was associated with increases in MSE (showing that an increase in complexity is good for memory), aging was ALSO associated with an increase in MSE, which seems paradoxical given that memory performance typically decreases and indeed decreased with age in the present study (Table 1). The authors suggest some speculative reasons for this paradox on pp. 10-11. The findings on age-related changes in the DMN in response to memory encoding seem to be quite mixed as far as I have read. The first idea appeared to be that aging was associated with ‘reduced deactivation’ of the DMN, but that has been challenged recently (Maillet & Schacter, 2016, TiCS). Given the importance of the topic the present article seems well worth reporting, and I support publication in Entropy.

We thank the reviewer for this insight and have added another possible interpretation for the age differences. Specifically, we now state, “A slightly different perspective is that older adults may rely more on processes subserved by the DMN generally and, depending on the context, this over-reliance might aid or hinder memory (Maillet & Schacter, 2006). Indeed, older adults often rely more on prior knowledge and schemas than younger adults (Baltes et al., 1984). Moreover, processing such schemas has been associated with connectivity between the medial temporal lobes and brain regions within the DMN such as the medial prefrontal cortex in similar post-encoding paradigm, at least in young adults (van Kesteren et al., 2010, 2014). Thus, the present findings are consistent with the notion that middle-aged and older adults rely more on the DMN than younger adults (Maillet & Schacter, 2006).”

My only comments are for clarification. For example, I would welcome a few statements about how the MSE measure relates to the general notions of activation and deactivation, and also to the issues of connectivity within and between networks. Also something more on the specific time scales sampled in the MSE analysis, and how these might reasonably relate to possible time scales of consolidation.

We thank the reviewer for their questions, which both highlights the novelty of using MSE approaches and reveals what research is still needed to fully understand how the analyses relate to other measures of brain functioning and cognition. Unfortunately, we are not aware of any studies that have compared MSE to more traditional measures of task-related activation and deactivation in fMRI. One reason is that one needs many time points to estimate MSE, which cannot be calculated for individual trials in an event-related fMRI design. While a blocked design might overcome this obstacle if the blocks were long enough, there still remains a dearth of studies investigating task-related MSE in fMRI. Thus, most of the research has been focused on resting-state MSE. We have used a similar network complexity approach (McDonough & Nashiro, 2014) using resting-state fMRI with an unusually short TR of .72 seconds and long duration (1200 seconds), which allowed a great characterization of fine and coarse time scales within the BOLD signal, thus achieving 25 time scales. When correlating network complexity and MSE across four different higher order networks (including the DMN), we found that greater MSE at fine time scales (around 1-3) was associated with lower functional connectivity within that network whereas greater MSE at mid to coarse time scales (10-25) was associated with greater functional connectivity within that network. Note that because of the different TR and number of time points compared to the current study, it is difficult to directly compare time scale between studies. While those findings speak to the relationship between MSE and within-network connectivity, they do not speak to between-network connectivity. Some research calculating MSE using EEG has found that fine time scales might represent local-connectivity and coarse time scales might represent distributed connectivity (e.g., between network or between hemisphere) (McIntosh et al., 2010; Mizuno et al., 2010; Vakorin et al., 2011). We have now highlighted many of these points as clarifications needed for future research in the Conclusion section.

Participants. Could we have a little more about how samples were recruited? eg were young adults college students? Recruited via advertisements? Were older adults in some sort of health program (line 132)?

We now added that “All participants were recruited from the Tuscaloosa and Birmingham areas within Alabama through word of mouth, flyers, Facebook ads, and newsletters.” Because the age range started at 20, most of the students had graduated college, but some were currently attending the local university. However, we did not recruit through psychology courses as in many other psychology studies, having a slightly more heterogenous sample of young adults. Older adults were not in a health program. The Alabama Brain Study on Risk for Dementia is simply the name of the study.

Procedure. Please include details of a) number of trials (not mentioned until line 394!), b) timing of each encoding trial – eg 4 per minute? Also the interval between study trials. With regard to the memory test (lines 163-168), was each test composed of 1 target and 4 lures – all previously seen? If so, does “never seen” refer to one possible response option, despite the point that the test was forced choice? Was the test paced or self-paced? Small but important points!

We thank the reviewer for pointing out these missing aspects of the procedure. We now state, “After the first resting-state scan, participants studied 64 pairs of pictures for the memory encoding task over two 8 minute sessions. The pictures consisted of either a face-object or face-scene pair for 3 seconds. After viewing each pair, participants were given 2.16 seconds to predict how likely they would remember the pair on a later memory test on a 3-point scale corresponding to likely, maybe, or unlikely. The inter-trial interval ranged from 1.72–17.20 seconds. The memory test consisted of a five-alternative-choice test in which participants were presented with a previously seen face and were asked to choose the object or scene that was previously paired with the face from five options: two objects, two scenes, and “never seen”. Of the four possible picture choices, one was a target and three were lures. Because all options were previously seen, participants needed to rely on recollection processes to answer correctly. The “never seen” option specifically referred to not remembering the face, thus precluding participants from making a correct response without complete guessing. Each of the 64 trials was presented for 5.16 seconds with intertrial intervals that ranged from 1.72-10.32 seconds. The memory test was divided into two runs lasting for 5 minutes.”